# ANGLE-DFQ: ANGLE AWARE DATA FREE QUANTIZATION

## ABSTRACT

Data free quantization of neural networks is a practical necessity as access to training data in many situations is restricted due to privacy, proprietary concerns, or memory issues. We introduce a data free weight rounding algorithm for Deep Neural Networks (DNNs) that does not require any training data, synthetic data generation, fine-tuning, or even batch norm statistics. Instead, our approach focuses on preserving the direction of weight vectors during quantization. We demonstrate that traditional weight rounding techniques, that round weights to the nearest quantized level, can result in large angles between the full-precision weight vectors and the quantized weight vectors, particularly under coarse quantization regimes. For a large class of high-dimensional weight vectors in DNNs, this angle error can approach 90 degrees. By minimizing this angle error, we significantly improve top-1 accuracy in quantized DNNs. We analytically derive the angle-minimizing rounding boundaries for ternary quantization under the assumption of Gaussian weights. Then, leaving the Gaussian assumption behind, we propose a greedy data-free quantization method based on the cosine similarity between the full-precision weight vectors and the quantized weight vectors. Our approach consistently outperforms existing state-of-the-art data-free quantization techniques and, in several cases, surpasses even data-dependent methods on well-established models such as ResNet-18, VGG-16, and AlexNet with aggressive quantization levels of 3 to 6 bits on the ImageNet dataset. Code will be made available at time of publication.

## 1 INTRODUCTION

Deep Neural Networks (DNNs) excel at many computer vision tasks. However, deploying these models on resource-constrained devices poses significant challenges due to their computational and storage requirements. Quantization is one promising approach to tackle these challenges. There are two common approaches to quantization, quantization aware training (QAT) and post training quantization (PTQ). QAT trains the model from scratch using the quantized weights and activation. Many works have shown the effectiveness of QAT (Hubara et al., (2016), Esser et al., (2019), Courbariaux et al., (2015), Rastegari et al.,(2016), Choi et al., (2015), Judd et al., (2015)). While these approaches hold promise, they are not always feasible in practical application areas. Accessing the original training data may not always be possible due a number or reasons including its size, privacy concerns, proprietary nature of the data, etc.

Post training quantization (PTQ) is particularly important as it allows users to deploy models in memory constrained environments without access to the entire original training data set. Yet many post training quantization methods are still dependent on a small subset of training data for calibration. Numerous papers have shown great results using quantization methods that rely on a small amount of training data for calibration (Nagal et al.,(2020), Hubara et al., (2021), Choukroun et al., (2019), Migacz et al., (2017), Lin et al., (2016), Han et al., (2015)). Yet there are still many situations where even accessing small amounts of data is impossible. For this reason industry has largely focused on model quantization schemes that do not require access to the training data for fine-tuning ( Nagel et al., (2019), Zhao et al., (2019)). Data Free quantization methods are therefore very important.

Recently, a new method of data free quantization was introduced- synthetic data generation. This allows quantization aware training without any training data. Several papers have shown impressive results in Qzero, DSG, ect (Cai et al., 2020, Qin et all 2023). However as noted in papers publishing new methods without data generation such as Squant and UDFC (Guo et al., 2022, Bai et al., 2023) generating the synthetic samples introduces extra computational costs, is complex, and depends on the availability of BN layers. The trade offs between these varied methods at times makes fair comparison difficult. The authors of DFQ (Nagel et al., 2018) previously proposed 4 levels of practical quantization applicability. Level 1, no training data and no back-propagation required. Level 2, requires data but no back-propagation. Level 3, requires data and back-propagation and works for any model. Level 4, requires data and back-propagation but only works for specific models. Now we propose an additional quantization level 0 that does not require any training of synthetic data nor back-propagation. In this paper we present a method that meets the level 0 quantization standard.

With the exception of Zhang et al., (2019) there is a dearth of data-free methods that consider quantization based angle errors. Data free weight quantization techniques have generally tried to minimize the quantization error by taking the MSE optimization approach which leads to a round-to-nearest quantization scheme. In this work we propose an alternative view. We show that at low bit implementations, quantization causes a large angle between the full precision weight vector associated with a single neuron and its rounded counterpart. This angular error changes the decision boundary and associated input space for which the Relu nonlinearity turns ON/OFF. Therefore, it has a large effect on the accuracy of the neural network. We propose a greedy algorithm to greatly reduce the angle error associated with course weight quantization. Below the contributions of this paper are summarized.

- We show that for a large class of weight vectors common in DNNs, round to nearest quantization can lead to very large angle errors, i.e. the angle between the full precision and the quantized weight vectors. High dimensional weight vectors, with the majority of weights near zero, can generate angle errors close to 90 degrees. In particular, we show that in the limit, the angle error can tend to 90 degrees under certain conditions. We will provide conditions under which large angles can occur under conventional round to nearest quantization and provide some illustrative examples from popular DNNs.

- We analytically derive the optimum rounding threshold for minimizing the angle error for ternary weight quantization (weights $\in \{-1, 0, 1\}$ ), assuming the weights have a Gaussian distribution.' We show that the rounding threshold for minimizing the angle error is much smaller than the round to nearest threshold (0.5) and depends on the distribution of the weights.

- We introduce a data free greedy algorithm for weight rounding that drastically reduces the angular error associated with weight quantization. This algorithm works for any word length implementation and makes no assumptions on the underlying distribution of the weights.

- Using our proposed rounding algorithm we show significant top-1 accuracy boosts on two well benchmarked models AlexNet, VGG16, and Resnet-18 on the imagenet dataset. These results highlight the importance of the angular error and its effect on model accuracy.

## 2 RELATED WORK

The near endless applications of effective edge AI have motivated a plethora of works in the DNN model quantization area. The problem of model compression for deployment in resource constrained environments is well known. Post-training-quantization (PTQ) methods had remarkable success in using fine-tuning with a small amount of training data to aggressively quantized pre-trained models. In this section we will discuss previous works that are relevant to our proposed method. We focus on works that present results on very course quantization of weights to below 8 bits.

Lybrand  Saab (2021) presented a Greedy Path-Following Quantization (GPFQ) that employed a deterministic quantization of the model layers in an iterative fashion without requiring a complex re-training. Zhang et al.,(2023) later improved and generalized the GPFQ method. This method showed near full precision accuracy results for weight quantization as low as 3 bits on the VGG16

and Alexnet models on Imagenet. While their results are very impressive they still rely on a small calibration set of training images to implement their quantization.

Recently, there has been a renewed focus on data free quantization techniques for Deep Neural networks (DNNs) that do not require any access to the original data set. These techniques are important when the underlying training data is private, proprietary, or difficult to process. This is an additional step beyond post training quantization (PTQ), as the latter sometimes still uses a small calibration subset of the original dataset to fine-tune the quantized model. Banner et al.,(2018) introduced a novel technique that exploited per channel quantization to achieve 4 bit quantization of DNNs with only small top-1 accuracy loss. This method was further notable as it did not require training data to fine-tune the quantized model weights. Although it uses some training data to determine clipping values for the activation, it a data-free method in terms of weight quantization. Nagel et al., (2019) proposed a data free quantization method that equalized the weight ranges across in a DNN to reduce quantization error bias. This method improved top-1 accuracy of quantized methods on well known DNNs without relying on a calibration subset of the original training data. While each of the above mentioned data-free quantization methods have unique contributions (per channel quantization, clipping, weight equalization, ect.) one commonality is that many employ a round to the nearest integer, or signed integer, quantization. These methods do not consider the angular error that quantization creates between a weight vector and its quantized counter part.

There are to the best of our knowledge relatively few data-free PTQ methods that consider the angle error. In the area of QAT binary quantization, Anderson et al. (2017) showed that the angle between a random vector (from a standard normal distribution) and its binarized counter part converges to 37 degrees as the dimension of the vector goes to infinity. Zhang et al., (2019) introduced a data-free PTQ method called Target None Re-training Ternary (TNT). The algorithm was of complexity $O(N \log N)$ and experiments were performed for a uniform and a gaussian distribution of the weights in a high dimensional vector. The experiments showed that for very long vectors with gaussian distribution cosine similarity was approximately 0.9, while there was significant uncertainty in the range of cosine similarity for shorter vectors. Using simulation experiments, the authors also investigated the optimal number of non-zero entries in the vector in order to obtain this maximum cosine similarity and concluded that the distribution and the number of non-zero elements have a large impact on the achievable cosine similarity. In the subsection on ternary quantization, our paper will revisit these topics in an analytical manner where cosine similarity is analyzed using the quantization boundary rather than the number of nonzero vector entries. Further this work provides an angle aware data free quantization (Angle-DFQ) method that extends beyond the ternary case and can be implemented for any bit width without any assumption on the underlying distribution of the weights.

## 3 METHODOLOGY

In this section we motivate the need to reduce quantization angle errors and offer a detailed explanation of our method. We show how quantization induced angle errors affect DDNs at the neuron level in section 3.1. In section 3.2 we show that significant quantization induced angle errors can occur for a large class of vectors commonly found in popular DNNs. In section 3.3 we present the illustrative special case of tertiary weight rounding (weights $\in \{-1, 0, 1\}$), where the rounding boundary for minimizing the decision plane angle error is much smaller than the round to the nearest value of 0.5. In section 3.4 we present our data free greedy weight rounding method that greatly reduces the angle error associated with course weight quantization.

### 3.1 ANGLE ERRORS AT THE NEURON LEVEL

Let us consider a single neuron with weight vector $\mathbf{v}$ and input vector $\mathbf{x}$. Then the operation of a neuron is modeled by the ReLU non-linearity in the following manner:

$$\text{Output} = \text{ReLU}(\mathbf{x} \cdot \mathbf{v} + b)$$

where $b$ is a scalar, i.e., the bias. If the argument of the ReLU nonlinearity is positive, the ReLU is ON and produces its argument as its output, otherwise it is OFF and produces a zero output. The relationship between $\mathbf{v}$, $\mathbf{x}$, and the ReLU output is shown for the 2-D case for $b = 0$ in Figure 1.

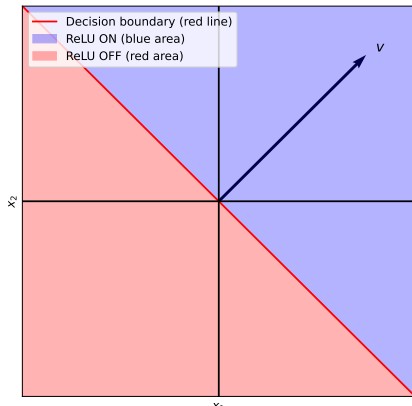 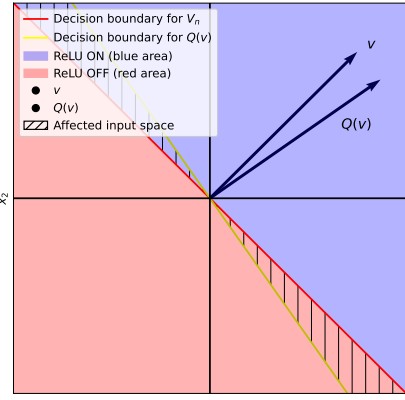

Figure 1: (Left) The weight vector $\mathbf{v}$ and its associated decision boundary determine the input space for the ReLU 'ON/OFF' result. (Right) The quantization-induced angle error of weight vector $\mathbf{v}$ changes the associated input space for the ReLU 'ON/OFF' result.

Any input vector in the red region creates a negative inner product result that would be zeroed out by the ReLU. Any input vector on the red line would be perpendicular to the $\mathbf{v}$ vector, resulting in a inner product of zero. The red line demarks the region between the ReLU 'on' input space and the ReLU 'off' input space and is always perpendicular to the weight vector $\mathbf{v}$ because of the inner product definition. We refer to this red line as the hyperplane decision boundary.

Quantization changes the direction of the weight vector, so there exists an angle $\phi$ between $\mathbf{v}$ and its quantized counterpart $\mathbf{Q}(\mathbf{v})$. This angle error changes the input space for a positive inner product result, i.e., the ReLU is 'ON'. This effect is shown in Figure 1

### 3.2 LARGE QUANTIZATION INDUCED ANGLE ERRORS

Consider a weight vector $\mathrm{v} \in \mathbb{R}^N$ associated with a single neuron. Lets define two vectors, $\mathbf{u}$ and $\mathbf{w}$ where $\mathbf{u}, \mathbf{w} \in \mathbb{R}^N$ and

$$\mathbf{u} = \begin{pmatrix} \epsilon_1 \\ \epsilon_2 \\ \vdots \\ \epsilon_{N-p} \\ 0 \\ \vdots \\ 0 \end{pmatrix}, \qquad \mathbf{w} = \begin{pmatrix} 0 \\ \vdots \\ 0 \\ w_{N-p+1} \\ \vdots \\ w_N \end{pmatrix},$$

$$|\epsilon_i| < \frac{1}{2}, \qquad |w_i| \geq \frac{1}{2}$$

Further assume the entries $\epsilon_i$ to come from a distribution that is symmetric around zero (i.e., zero mean) and to have a variance of $\sigma^2$. Assume the elements of $\mathbf{w}$ to be fixed as well as their number, i.e., $p$. We can now define the entire weight vector $\mathbf{v}$ to be:

$$\mathbf{v} = \mathbf{u} + \mathbf{w} = \begin{pmatrix} \epsilon_1 \\ \epsilon_2 \\ \vdots \\ \epsilon_{N-p} \\ w_{N-p+1} \\ \vdots \\ w_N \end{pmatrix} \qquad (1)$$

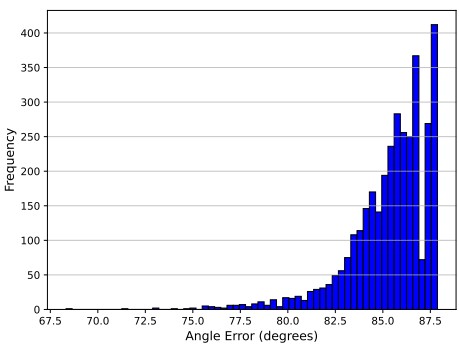 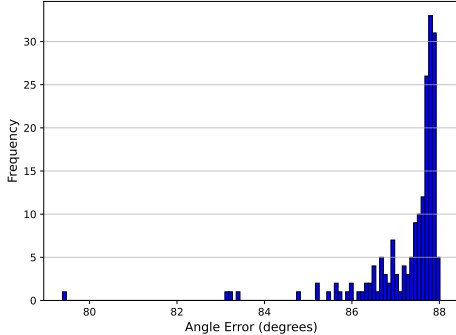

Figure 2: Histogram of the angle errors of all weight vectors in classifier[1], the largest layer of AlexNet (left) and classifier[0], the largest layer of VGG16 (right), under round to nearest ternary quantization(weights $\in \{-1, 0, 1\}$). The average angles for the layers are 85.35 degrees (AlexNet) and 87.30 degrees (VGG16).

Note that while elements in $\mathbf{v}$ are partially ordered, the order of elements in a vector has no effect on the angle between this vector and its quantized version, since neither affects the 2-norm of either vector nor their inner product.

Defining $Q(\mathbf{v})$ as the quantized vector of $\mathbf{v}$ where quantization is performed elementwise, i.e.,

$$Q(\mathbf{v}) : \mathbb{R}^N \to \mathbb{Z}^N$$

$$Q(\mathbf{v}) = \begin{pmatrix} q(v_1) \\ q(v_2) \\ \vdots \\ q(v_N) \end{pmatrix}$$

with $q(\cdot)$ being an odd function and $\mathbb{Z}$ is the set of integers. Using magnitude rounding (rounding to nearest), we have equation 1 below:

$$Q(\mathbf{v}) = \begin{pmatrix} 0 \\ \vdots \\ 0 \\ Q(w_{N-p+1}) \\ \vdots \\ Q(w_N) \end{pmatrix} = Q(\mathbf{w}) \quad (2)$$

**Theorem 1:** Let $\mathbf{v}$ be defined as in (1), Then the angle $\phi$ between $\mathbf{v}$ and $Q(\mathbf{v})$ tends to $\frac{\pi}{2}$ for $N \to \infty$, i.e.,

$$\lim_{N \to \infty} \phi(N) = \frac{\pi}{2}$$

assuming $p$ and $w_i$, $i = N - p + 1, \ldots, N$ are fixed and the elements $\epsilon_i$ come from the same distribution with variance $\sigma^2$.

**Proof:** With

$$\cos \phi = \frac{\mathbf{v} \cdot Q(\mathbf{v})}{(\|\mathbf{v}\|_2)(\|Q(\mathbf{v})\|_2)}, \quad (3)$$

and using (1) and (2) we obtain:

$$\cos \phi = \frac{\sum_{i=N-p+1}^{N} w_i Q(w_i)}{\sqrt{\sum_{i=1}^{N-p} \epsilon_i^2 + \sum_{i=N-p+1}^{N} w_i^2} \cdot \sqrt{\sum_{i=N-p+1}^{N} Q(w_i)^2}} \tag{4}$$

Keeping $p$ and the $w_i$ fixed, we obtain with $N \to \infty$ :

$$\lim_{N \to \infty} \cos(\phi) = \lim_{N \to \infty} \frac{\sum_{i=N-p+1}^{N} w_i Q(w_i)}{\sqrt{(N-p)\left(\frac{\sum_{i=1}^{N-p} \epsilon_i^2}{N-p}\right) + \sum_{i=N-p+1}^{N} w_i^2} \cdot \sqrt{\sum_{i=N-p+1}^{N} Q(w_i)^2}}$$

$$= \lim_{N \to \infty} \frac{\sum_{i=N-p+1}^{N} w_i Q(w_i)}{\sqrt{(N-p)\sigma^2 + \sum_{i=N-p+1}^{N} w_i^2} \cdot \sqrt{\sum_{i=N-p+1}^{N} Q(w_i)^2}} = 0, \text{ and hence } \phi \to 90° \text{ for } N \to \infty.$$

This shows that as N approaches infinity, the angle $\phi$ between v and Q(v) asymptotically tends to 90°, assuming that a constant and finite number of vector entries in v are larger than $\frac{1}{2}$ while the number of entries smaller than $\frac{1}{2}$ in magnitude tends to infinity.

**Remark:** The conditions on the entries of vector **w** can be shown to be somewhat conservative, i.e., they are not necessary conditions for the theorem to hold. In fact, the angle error can approach $\frac{\pi}{2}$ for $N \to \infty$ even if $p$ also tends to infinity, but at a slower rate than $\sqrt{N}$ and with the condition that its elements are bounded. For example, for $N \to \infty$, $p$ can grow with $\log N$ and the angle $\phi$ would still tend to $\frac{\pi}{2}$ if elements of **w** remain bounded. On the other hand, if $p$ grows with $cN$, $c$ being a small positive real number between 0 and 1, the error angle will not converge to $\frac{\pi}{2}$ for $N \to \infty$. In practice, this means that when weight vectors in DNNs become longer, if the ratio of entries larger than (the round to nearest threshold) $\frac{1}{2}$ to those smaller than $\frac{1}{2}$ is approximately constant, the asymptotic error angle will be less than $\frac{\pi}{2}$ and can be computed using equation (3) in the proof.

In other words, long vectors with narrow weight distributions around zero that also have a few large entries can produce large angles $\phi$. This theoretical result is consistent with the angle errors we observe when quantizing popular DNNs. In figure 2 we present a histogram of the angle errors of the weight vectors in the largest layer of AlexNet and VGG-16 under ternary round to the nearest quantization and report an average angle error of 85.35 and 87.30 degrees for repsective layers. In Resnet-18, the largest layer is the last convolutional layer which has an average angle error of 77.02 degrees under 2 bit round to nearest with the coresponding figure in appendix a2.

### 3.3 AN ANGLE MINIMIZING ROUNDING THRESHOLD FOR TERNARY QUANTIZATION

In this section we will analytically derive the angle minimizing rounding threshold for ternary quantization and show that rounding to nearest is non-optimal for minimizing the angle error, under the assumption of Gaussian distributed weights. We will further show that the angle minimizing rounding threshold approaches zero as the variance of the underlying distribution of the weights approaches zero. This depends on the characteristics of the pdf of the weights. The lessons learned about the angle minimizing rounding threshold in illustrative special case of ternary quantization (weights $\in \{-1, 0, 1\}$) provide insight for higher word length situations. In round to the nearest quantization a rounding threshold of magnitude 0.5 would be used. Below we show that the rounding threshold k that minimizes the quantization angle error is not 0.5 in the Gaussian case.

Assume a Gaussian weight distribution with a mean of zero and a variance of $\sigma^2$. Further assume there are only three q-levels, namely -1,0,and 1 and the weight vector to be of dimension $N \to \infty$. With an odd quantization non-linearity and rounding thresholds -k and k, we can derive the following equation on the angle error $\phi$.

$$\cos^2(\phi) = \frac{\left(\frac{1}{\pi}\right) e^{-\frac{k^2}{\sigma^2}}}{(1 - cdf[k])}$$

This expression was derived in the appendix A for the asymtotic case, i.e. $N \to \infty$. This is an expression for the angle error in terms of the variance of the weights and the rounding threshold k.

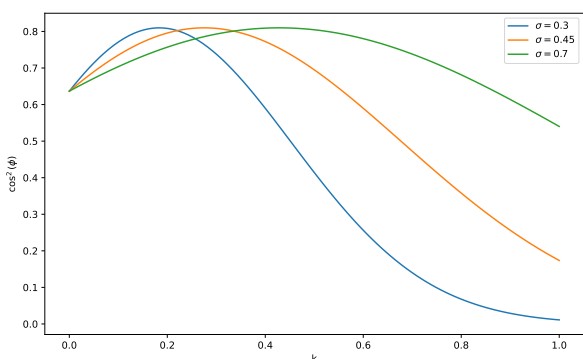

Figure 3: Plots of the cosine similarity squared $\cos^2(\phi)$ as a function of $k$, for different values of $\sigma$.

Given a variance for the weights we can now show that the optimum k value is generally not the round to nearest threshold of 0.5. Figure 3 explains the dependency of $\cos^2(\phi)$ as a function of the rounding boundary $k$, parameterized for three different values of $\sigma$. The locations of the function maxima depend on $k$, and are attained approximately at $k = \frac{5}{8}\sigma$. This maximum function value of $\cos^2(\phi)$ is approximately 0.81 and independent of $\sigma$ and it corresponds to a minimal attainable angle of $25.8°$. This clearly shows that for small $\sigma$ approaching zero, the rounding boundary $k$ will also approach zero. Also observe that $k = 0$ results in an angle of approximately $37°$ which is also independent of the value of $\sigma$, since all functions intersect at the point $k = 0$. Note that a rounding boundary of $k = 0$ corresponds to binary quantization, i.e., only the two quantization levels 1 and -1 are used.

The interesting special case of rounding boundary k= 0 implying a binary quantization was previously analyzed by Anderson et. al (2017). For k = 0 the above expression becomes:

$$\cos^2(\phi) = \frac{\left(\frac{1}{\pi}\right)e^0}{(1 - cdf[0])} = \frac{2}{\pi}$$

showing the angle error $\phi$ = 37 degrees in such a situation.

In this section we have derived an angle minimizing weight rounding threshold for ternary quantization in the Gaussian case. Our analytical result shows a maximum function value of $\cos^2(\phi) = 0.81$, independent of $\sigma$, and corresponds to the minimal attainable angle of $25.8°$. This matches simulation experiments by Zhang et al. (2019) that showed that for very long vectors with Gaussian distribution cosine similarity was approximately 0.9.

### 3.4 A Data-Free Greedy Weight Rounding Algorithm to Implement Angle-DFQ

In this section we present a data-free greedy weight rounding algorithm. We will discuss this method in the context of signed integer per layer weight rounding but it is also relevant to per channel implementations. We have already showed in section 3.2 how to do optimal rounding for in the special case of tertiary quantization where the weights arise from a normal distribution. Here we wish to show a more general method that extends beyond tertiary quantization and is applicable regardless of the underlying distribution. This greedy method aims to minimize the angle between $\mathbf{v}$ the full precision weight vector of a single neuron with n weights, and $\mathbf{Q(v)}$ the quantized $\mathbf{v}$. We round each element in the vector up or down depending on the effect on the angle error before moving to the next element. Once the layer is quantized we calculate the magnitude error factor $\mathbf{E_m}$ of each $\mathbf{Q(v)}$ introduced by the angle aware weight rounding.

$$E_m = \frac{\|Q(v)\|}{\|v\|}$$

We then multiply all out going weights from the neuron associated with the particular $\mathbf{Q(v)}$ with the reciprocal. If due to the structure of the network its difficult to track the down stream weights,

divide all weights in the layer by the average $\mathbf{E_m}$. The algorithm is described precisely in algorithm 1.

---

**Algorithm 1: Angle-DFQ**

---

1. Choose a layer in a neural network to quantize to bit width $m$.

2. Scale the layer to fit within the integer range determined by $m$ (for per layer quantization).

3. For each weight vector $v$ in the scaled layer:

    (a) Initialize $Q(v)$ with the first two elements of $v$.

    (b) Round the elements in such a way that minimizes the angle $\phi$ between the first two elements of $Q(v)$ and the first two elements of $v$ by computing the angle for all four rounding options.

    (c) For $i = 2$ to the length of $v$:

        i. Round the $i$-th element of $v_n$ up and append it to $Q(v)$.

        ii. Compute and record the angle between $Q(v)$ and the first $i$ elements of $v$.

        iii. Now remove the $i$-th element of $Q(v)$ and append the rounded down $i$-th element of $v$ to $Q(v)$ and again compute the angle between $Q(v)$ and the first $i$ elements of $v$.

        iv. Select the rounding that minimizes the angle and append the rounded element to the quantized weight to $Q(v)$.

4. Compute the magnitude error for the quantized vector:

    (a) Calculate $E_m = \frac{\|Q(v)\|}{\|v\|}$ for each vector to get the magnitude error introduced by Angle-DFQ.

    (b) Then for each neuron multiply all outgoing weights by the reciprocal of the magnitude error, $\frac{1}{E_m}$ to correct the magnitude error, or divide all weights by the average $\mathbf{E_m}$ across the whole layer if down stream are difficult to track

---

In the next section we will apply this method to several well bench marked models on the ImageNet Dataset and show near full precision accuracy at very low bit implementations. We will now highlight some implementation details, starting with quantization granularity. The trade-offs between per layer and per channel quantization are well known. Per channel quantization yields higher accuracy but requires more overhead and can be more difficult to implement across different hardware (Gholami et al., 2022,Nagel et al. 2021). To have a fair comparison we will follow convention and compare to other techniques that are at least per layer in granularity. For activation quantization quantization (per layer) we use the same data free approach as in DFQ (Nagel et al. 2019) where the range is set with the batch norm statistics. If the model does not have batch norm layers then a small amount of data is used (200 training images) to collect statistics for the ranges of the activation quantization. This does not violate our data free policy as we are presenting a data free weight rounding algorithm. All novel contributions in this work pertain strictly to weight rounding.

### 3.5 MIXED PRECISION QUANTIZATION

Mixed precision quantization is a common technique used by many popular quantization methods. From the methods we compare against, for example, MSE (Banner et al. 2018) for example uses mixed precision even at the per channel level, and GPFQ by not quantizing their last layers on some implementations. We can further take advantage of our knowledge of quantization induced angle error for mixed precision implementations. Since we know from our proof in section 3.2 that high dimensional weight vectors have the largest angle error we will apply our Angle-DFQ to these layers where angle error dominates. Since these layers have the most high dimensional vectors they are the largest layers in models in terms of total numbers of weights. We quantize them to very course bit widths since we know that the Angle-DFQ can correct the extreme angle error. For the remaining layers of the models where angle error does not heavily dominate, round-to-nearest quantization is employed. The per layer bit allocation is assigned in such a way as to equalize the angle error across the layers. We detail the precise bit allocations for each layer and each model in Appendix A.2

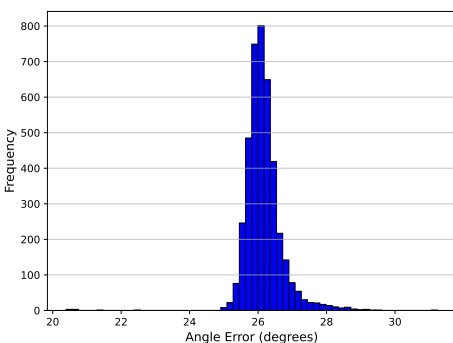 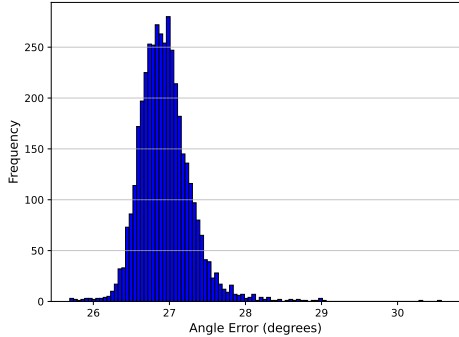

Figure 4: (Left) Histogram of the angle errors of all weight vectors in classifier[1], the first fully connected layer of AlexNet for ternary Angle-DFQ weight quantization (weights $\in \{-1, 0, 1\}$). The average angle for the layer is 26.16 degrees, much lower than the earlier RTN result. (Right) Histogram of the angle errors of all weight vectors in classifier[0], the first fully connected layer of VGG16 for ternary Angle-DFQ weight quantization (weights $\in \{-1, 0, 1\}$). The average angle for the layer is 26.94 degrees, much lower than the earlier RTN result.

and report our weighted average of total bits per weight for the given DNNs - these are rounded for simplicity in table 1.

## 4 EXPERIMENTS

In this section we showcase the significant impact that reducing quantization based angle errors can have on top-1 accuracy of Deep Neural Netowrks. We present results on three popular and well bench marked models: Resnet-18 (He et al. 2016), AlexNet (Krizhevsky et al. 2012), and VGG16 (Simonyan Zisserman, 2014) on the ImageNet dataset.

### 4.1 ANGLE-DFQ AND ANGLE ERROR REDUCTION

In section 3.1 we discussed the large angle errors that could occur from round to the nearest (RTN) quantization. We presented histograms in figure 2 showing the angle errors in the largest layers of AlexNet and VGG-16 under ternary round to the nearest weight quantization. A similar result can be found for Resnet-18 in the appendix for the 2 bit asymmetric case. We showed that the largest layer had an average angle error of 85.35 degrees in AlexNet, an average angle error of 87.30 degrees in VGG-16, and an average angle error of 77.01 degrees in Resnet-18. After applying our Angle-DFQ to these layers we show large improvements in the average angle error. Figure 4 presents a histogram of the angle errors in the largest layer of AlexNet, VGG-16 after Angle-DFQ showing an average angle error of only 26.16 and 26.94 respectively. Figure 5 in appendex shows the largest layer of Resnet-18 after Angle-DFQ has an angle error of only 29.49 degrees These results are close to the theoretical minimum angle error of 25.8 for ternary quantization in the Gaussian case established in section 3.3. These results show that Angle- DFQ is able to significantly reduce the angle error due to quantization in DNNs. In the next section we will show the Top-1 accuracy boosts derived from this angle error correction.

### 4.2 ANGLE-DFQ ON POPULAR DNNS

Table 1 shows our results along with other several other notable methods on low bit weight quantization. Note that where the † is used for the GPFQ method in table 1 it indicates that the last layer is left in full precision. In the case of the TNT algorithm by Zhang et al., (2019) they leave both the first and the last layer in full precision and thus we have reported a mixed precision average bit width in Table 1. We distinguish which methods are data-free and which are not for weight rounding. For the purposes of weight rounding we consider MSE data free as discussed previously. To show the flexibilty of the Angle-DFQ method we report results for unsymmetrical integer quantization for Resnet-18 and symmetrical integer quantization for AlexNet and VGG-16. Obviously, we can not compete against methods that use data generation or very fine granularity quantization

| Model | Method | $\sim D$ | $\sim PL$ | Bits(w/a) | Accuracy (top-1) | Reference Acc |
|---|---|---|---|---|---|---|
| ResNet-18 | Angle-DFQ (ours) | ✓ | ✓ | 6/6 | **66.81%** | 69.7% |
| | DFQ | ✓ | ✓ | 6/6 | 66.30% | 69.7% |
| | Krishnamoorthi (per layer) | ✓ | ✓ | 6/6 | 63.9% | 69.7% |
| | Squant | ✓ | ✗ | 6/6 | 70.74% | 71.47% |
| | Qzero | ✗ | ✓ | 6/6 | 71.3% | 71.47% |
| | UDFC | ✓ | ✗ | 6/6 | 72.76% | 71.47% |
| VGG-16 | Angle-DFQ (ours) | ✓ | ✓ | 2.81/32 | **71.11%** | 71.59% |
| | TNT | ✓ | ✓ | 2.89/32 | 64.4% | 71.59% |
| | Angle-DFQ (ours) | ✓ | ✓ | 3/32 | **71.16%** | 71.59% |
| | Angle-DFQ (ours) | ✓ | ✓ | 3/8 | **71.01%** | 71.59% |
| | MSE | ✓ | ✗ | 3/8 | 69.5% | 71.59% |
| | GPFQ | ✗ | ✓ | 3/32 | 70.24% | 71.59% |
| | GPFQ | ✗ | ✓ | 5/32 | 70.96% | 71.59% |
| AlexNet | Angle-DFQ (ours) | ✓ | ✓ | 3/8 | **54.06%** | 56.52% |
| | GPFQ | ✗ | ✓ | 3/32 | 53.22% | 56.52% |
| | Angle-DFQ (ours) | ✓ | ✓ | 4/32 | **55.74%** | 56.52% |
| | OMSE | ✗ | ✗ | 4/32 | 55.52% | 56.52% |
| | GPFQ | ✗ | ✓ | 4/32 | 55.15% | 56.52% |
| | GPFQ$^\dagger$ | ✗ | ✓ | 4/32 | 55.51% | 56.52% |
| | Angle-DFQ (ours) | ✓ | ✓ | 5/32 | **56.07%** | 56.52% |
| | GPFQ | ✗ | ✓ | 5/32 | 55.67% | 56.52% |
| | GPFQ$^\dagger$ | ✗ | ✓ | 5/32 | 55.94% | 56.52% |

Table 1: Top-1 accuracy results for different models and quantization approaches on ImageNet. (D: synthetic/training data free, PL: at least per layer quantization implementation)

but as discussed in the introduction the implementation trade-offs for such methods are well known. Nonetheless, we list some such methods in Table 1 for completeness.

Our Angle-DFQ method shows state of the art results superior to other data-free quantization methods on the Resnet-18, VGG-16 and AlexNet datasets for the reported bit widths in table 1. We further show accuracy improvements superior to the data-dependent method GPFQ on VGG-16 for bit widths all reported bit widths (3 to 5) and to MSE (Banner, 2018) on the 3 bit weight 8 bit activation case. Moreover, we show results superior to GPFQ on AlexNet even for the 4 and 5 bit case when they do not quantize the last layer (GPFQ$^\dagger$). Angle-DFQ also out preforms the data dependent OMSE method on the 4 bit case for Alexnet. These results showcasing the importance of correcting angle errors in the quantization process.

## 5 CONCLUSION

This paper presents an in-depth analysis of quantization effects on the angle of the weight vector and equivalently the angle of the decision hyperplane of a neuron. It is shown that under certain conditions the angle between the full precision and the quantized weight vector can approach $90°$, which corresponds to almost half of the input space being classified incorrectly by the ReLU non-linearity. It is also shown that the error angle minimizing quantization boundary is not $n + \frac{1}{2}$ as is the default method when minimizing the 2-norm of the error vector under the round to the nearest method. Using the case of ternary quantization, it is shown that the optimal quantization boundary depends on the distribution of weights and can be close to a quantization point, i.e., zero in the case of tertiary quantization.

Armed with this theoretical foundation, we introduced Angle-DFQ; a data-free quantization method that greatly boosts quantized model accuracy without the need for data or fine-tuning. While the Angle-DFQ algorithm is not guaranteed to always find the optimal quantized weight vector, it shows low complexity and exhibits accuracy improvements for weight quantization in Resnet-18, AlexNet and VGG-16. The simplicity and straight forward nature of the Angle-DFQ method is a further advantage for adaptation of this work in industry. The Angle-DFQ technique is well fitted for deploying models in the memory constrained environments required in many edge AI applications.

# 6 ETHICS STATEMENT

This paper presents work whose goal is to advance the field of Machine Learning. There are many potential societal consequences of our work. We accept and agree with the ICLR code of ethics.

# 7 REPRODUCIBILITY

We agree that reproducibility is an important part of scientific research. To further this end we have been clear and obvious with each step of our implementation. We provided a full proof of our theoretical results in appendix A1 and specific tables in appendix A2 that show the exact bit allocation of our mixed precision implementations.

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

# A APPENDIX

## A.1 DERIVATION FROM SECTION 3.3: AN ANGLE MINIMIZING ROUNDING THRESHOLD FOR TERNARY QUANTIZATION

Consider a weight vector v in a neural network. Let $v \in \mathbb{R}^N$ and $Q(v) \in \mathbb{Z}^N$, with $\mathbb{R}$ being the set of all real numbers and $\mathbb{Z}$ the set of all integers. Define a mapping $Q : v \to Q(v)$ such that $Q(v) = (q(v_1), q(v_2), \ldots, q(v_N))$, where $v = (v_1, v_2, \ldots, v_N)$. In other words, quantization $q()$ is done elementwise where $q$ is: $\mathbb{R} \to \mathbb{Z}$.

Furthermore, let $q(v_i)$ be an odd function, that is, $q(-v_i) = -q(v_i)$. Therefore, we have: $(-v_i) \cdot q(-v_i) = (-v_i) \cdot (-q(v_i)) = v_i \cdot q(v_i)$.

Denoting $|v|$ as $|v| = (|v_1|, |v_2|, \ldots, |v_N|)$, it becomes clear that: $V \cdot Q(v) = |v| \cdot Q(|v|)$,

This implies that the inner product between $v$ and $Q(v)$ is independent of the sign of vector elements.

Also note that the 2-norm of any vector is independent of the sign of the vector entries, i.e., $||v||_2 = |||v|||_2$

Since,

$$\cos \phi = \frac{v \cdot Q(v)}{(||v||_2)(||Q(v)||_2)},$$

we can also write

$$\cos\phi = \frac{|v| \cdot Q(|v|)}{(|||v|||_2)(||Q(|v|)||_2)}$$

i.e., the angle $\phi$ between $v$ and $Q(v)$ does not change if all elements of $v$ are replaced by their absolute value.

Considering the asymptotic case for N to infinity we now transition from the discrete to the continuous case.

Let the elements of $v$ be distributed according to $pdf(v_i)$, where $pdf$ is the probability density function that describes the probability of vector elements $v_i$ occurring.

Assuming that

$$pdf(v_i) = pdf(-v_i)$$

i.e., a symmetric pdf with respect to zero (0), we have

$$pdf(|v_i|) = \begin{cases} 2 \cdot pdf(v_i) & \text{for } v_i \geq 0, \\ 0 & \text{for } v_i < 0. \end{cases}$$

Therefore, in the case of symmetric pdfs around zero, one can analyze the angle between $v$ and $Q(v)$ by analyzing the angle between $|v|$ and $Q(|v|)$ using $pdf(|v_i|)$.

Now consider ternary Quantization:

$$Q : \mathbb{R}^N \to \{-1, 0, +1\}^N$$

i.e., a quantizer with only 3 quantization levels.

Before proceeding further, we need to point out another property of the inner product between $Q(v)$ and $v$:

$$Q(v) \cdot v = Q(\hat{v}) \cdot \hat{v}$$

where $\hat{V}$ is generated by reordering the elements of $v$.

In fact, since $v$ and $\hat{v}$ have the same entries (just at different positions) $\|v\|_2 = \|\hat{v}\|_2$ also holds.

Now consider a vector $v$ with the following normal pdf:

$$pdf(v) = \frac{1}{\sqrt{2\pi}\sigma} e^{-\frac{v^2}{2\sigma^2}}$$

and therefore for the vector $|v|$, the pdf is given by:

$$pdf(|v_i|) = \begin{cases} \frac{2}{\sqrt{2\pi}\sigma} e^{-\frac{v_i^2}{2\sigma^2}} & \text{for } v_i \geq 0 \\ 0 & \text{for } v_i < 0 \end{cases}$$

(Note that $\phi(v, Q(v)) = \phi(|v|, Q(|v|))$ as shown above.)

We reorder the elements of $|v|$ in descending order:

$$|\hat{v}| = (\varepsilon_1, \ldots, \varepsilon_{\beta N}, \mu_1, \ldots, \mu_{(1-\beta)N})$$

where $\varepsilon_i \geq \varepsilon_{i+1}, \varepsilon_{\beta N} \geq \mu_1, \mu_i \geq \mu_{i+1}, \mu_{(1-\beta)N} \geq 0$ with $0 < \beta \leq 1$,

$$q(\varepsilon_i) = 1, \quad q(\mu_i) = 0$$

Therefore in $|\hat{v}|$, there are $\beta N$ elements that round to 1 and $(1 - \beta)N$ elements that round to zero.

Evaluating $|\hat{v}| \cdot Q(|\hat{v}|)$ we get:

$$\cos\phi = \frac{|\hat{v}| \cdot Q(|\hat{v}|)}{\||\hat{v}|\|_2 \cdot \|Q(|\hat{v}|)\|_2}$$

we obtain with the above equations:

$$\cos(\phi) = \frac{\sum_{i=1}^{\beta N} \epsilon_i}{\sqrt{\sum_{i=1}^{\beta N} \epsilon_i^2 + \sum_{i=1}^{(1-\beta)N} \mu_i^2} \sqrt{\beta N}}$$

With defining $\bar{\epsilon}$ as the mean of all $\epsilon_i$, we have:

$$\bar{\epsilon}\beta N = \sum_{i=1}^{\beta N} \epsilon_i$$

Therefore we obtain for $\cos^2(\phi)$:

$$\cos^2(\phi) = \frac{(\bar{\epsilon}\beta N)^2}{\left(\sum_{i=1}^{\beta N} \epsilon_i^2 + \sum_{i=1}^{(1-\beta)N} \mu_i^2\right)(\beta N)}$$

Using:

$$\sum_{i=1}^{N} v_i^2 = \left(\sum_{i=1}^{\beta N} \epsilon_i^2 + \sum_{i=1}^{(1-\beta)N} \mu_i^2\right)$$

and

$$\sigma^2 = \frac{\sum_{i=1}^{N} v_i^2}{N}$$

we obtain:

$$\cos^2(\phi) = \frac{\bar{\epsilon}^2 \beta}{\sigma^2}$$

Expressing $\beta$ in terms of k, where k is the rounding boundary:

$$\beta = \int_k^\infty \text{pdf}(|v_i|)\, dv_i = 2 \int_k^\infty \text{pdf}(v_i)\, dv_i$$

Now we will write $\epsilon$ in terms of k.

$$\bar{\epsilon} = \frac{\int_k^\infty v_i\, \text{pdf}(v_i)\, dv_i}{\int_k^\infty \text{pdf}(v_i)\, dv_i}$$

The expression for $\cos^2(\phi)$ will therefore become:

$$\cos^2(\phi) = \frac{\left(\int_k^\infty v_i\, \text{pdf}(v_i)\, dv_i\right)^2 \cdot \left(2\int_k^\infty \text{pdf}(v_i)\, dv_i\right)}{\left(\int_k^\infty \text{pdf}(v_i)\, dv_i\right)^2 \cdot \sigma^2} = 2\frac{\left(\int_k^\infty v_i\, \text{pdf}(v_i)\, dv_i\right)^2}{(cdf[\infty] - cdf[k]) \cdot \sigma^2}$$

Using the identity for normal PDFs

$$\int_k^\infty x\, \text{pdf}(x)\, dx =$$

$$= \left(\frac{\sigma^2}{\sqrt{2\pi}\sigma}\right) e^{-\frac{k^2}{2\sigma^2}}$$

we obtain:

$$\cos^2(\phi) = 2 \frac{\left(\frac{\sigma^4}{2\pi\sigma^2}\right) e^{-\frac{k^2}{\sigma^2}}}{(cdf[\infty] - cdf[k]) \cdot \sigma^2} = \frac{\left(\frac{1}{\pi}\right) e^{-\frac{k^2}{\sigma^2}}}{(1 - cdf[k])}$$

## A.2 MIXED PRECISION BIT ALLOCATION PER LAYER

The reported mixed precision bit widths represent a weighted average bit per weight in the DNN. These per layer bit widths are reported below. Some of these are rounded in the main text of the paper but we report the exact bit per weight average here. Note that the percentage of weights found in the first fully connnected layer is a majority of the total weights in the DNN for both AlexNet and VGG16.

Table 2: ResNet-18 bit allocation per layer for 6-bit case.

| Layer | 6-bit case | |
|---|---|---|
| layer4[1].conv2.weight.data | 2[†] | |
| layer4[1].conv1.weight.data | 6 | [†] Quantized with Angle-DFQ |
| layer4[0].conv2.weight.data | 6 | |
| All other layers | 8 | |

Table 3: Alexnet bit allocation per layer

| Layer | 2.92 bit case | 3.92 bit case | 4.96 bit case | |
|---|---|---|---|---|
| features[0].weight.data = | 8 | 9 | 11 | |
| features[3].weight.data = | 8 | 9 | 11 | |
| features[6].weight.data = | 8 | 9 | 11 | |
| features[8].weight.data = | 8 | 9 | 11 | [†]Quantized with |
| features[10].weight.data = | 8 | 9 | 11 | |
| classifier[1].weight.data = | 2[†] | 3[†] | 3[†] | |
| classifier[4].weight.data = | 3[†] | 4[†] | 7 | |
| classifier[6].weight.data = | 8 | 9 | 11 | |

Angle-DFQ

Table 4: VGG 16 bit allocation per layer

| Layer | 2.82 bit case | 2.95 bit case |
|---|---|---|
| features[0].weight.data = | 8 | 9 |
| features[2].weight.data = | 8 | 9 |
| features[5].weight.data = | 8 | 9 |
| features[7].weight.data = | 8 | 9 |
| features[10].weight.data = | 8 | 9 |
| features[12].weight.data = | 8 | 9 |
| features[14].weight.data = | 8 | 9 |
| features[17].weight.data = | 8 | 9 |
| features[19].weight.data = | 8 | 9 |
| features[21].weight.data = | 8 | 9 |
| features[24].weight.data = | 8 | 9 |
| features[26].weight.data = | 8 | 9 |
| features[28].weight.data = | 8 | 9 |
| classifier[0].weight.data = | $2^{\dagger}$ | $2^{\dagger}$ |
| classifier[3].weight.data = | $2^{\dagger}$ | $2^{\dagger}$ |
| classifier[6].weight.data = | 8 | 9 |

[†] Quantized with Angle-DFQ

## A.3 ANGLE ERROR OF RESNET-18

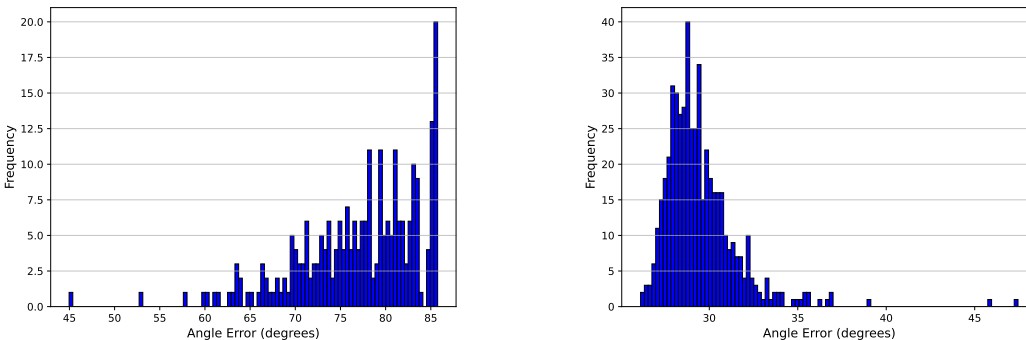

Figure 5: Angle error in layer4[1].conv2 of Resnet-18 before and after Angle-DFQ

## A.4 DATA AND MODELS

The models used (Resnet-18, AlexNet and VGG-16) are the pre-trained models from the PyTorch torchvision library. They have the BSD 3-Clause License.