# OpenReview forum: "Angle-DFQ: Angle aware data free quantization"
_ICLR.cc/2025/Conference — ICLR 2025 Conference Withdrawn Submission_

### Official Review · Reviewer_qNBm · 2024-10-22

**Soundness:** 1
**Presentation:** 1
**Contribution:** 2
**Rating:** 1
**Confidence:** 4

**Summary:**

This paper aims to establish a data-free post-training quantization algorithm for low-memory deployment of neural networks. The paper starts off with an analysis of the angle-errors of the quantized weight-vectors in a neural network, which can negatively affect the decision boundaries of the ReLU activation functions. Additionally, the paper solves the problem of the optimal decision boundary for ternary quantization. The paper then presents Angle-DFQ, an optimization algorithm that tries to greedily minimize the angle errors during quantization, and finally the papers evaluates their method on different neural networks from the computer vision community.

**Strengths:**

- The paper presents an intriguing approach to think about quantization in the term of angle-errors between weight vectors
- The solution to the ternary quantization problem presented in the paper is interesting and clear
- The proposed method is straightforward and easy to understand, and the motivation is made clear through earlier parts of the paper

**Weaknesses:**

### Presentation
- Spelling / Grammar issues ("due a number", "ect" instead of "etc";  "course weight quantization" instead of "**coarse** weight quantization";  misspelled proper nouns such as "Relu", "Alexnet", "Imagenet"; missing commas; ". ‘ We show", ...),
- Severe formatting errors (misplaced spaces, sometimes using equation environments and sometimes not for numbers,...), especially for references (References are not cited uniformly, f.e. when sources are cited in parentheses, the paper sometimes uses parentheses for the year, sometimes not; the reference on p1. Nagal et al 2020 does not even exist; "Qin et all 2023", etc).
- Presentation is poor at times: Table 1 is a low-resolution screenshot instead of a proper latex table, the equation under Eq. (4) is out of the bounding box of the text, Equations are numbered seemingly at random (and are non-uniformly referenced, f.e. "equation 1", "equation (3)", "(1)"), the first equation in section 1 has odd formatting for the constraints on the vectors.
- The derivations have some issues that make them harder to follow, f.e. in Appendix A p.13, the definition of the pdf should be for $|v_{i}| \geq 0$, not $v_{i} \geq 0$ (similarly for the other case), the symbol for $\varepsilon$ is switched mid-way through the derivation (from p.13 to p.14), ...

### Soundness
- Theorem 1 itself seems to have little relevance, as it only applies to continuous probability distributions whose support is restricted to $x < |\frac{1}{2}|$. Almost all practically used continuous probability distributions that are used to model neural network weights (such as zero-centered gaussians or laplace distributions) do _not_ share this property. The paper mentions in a remark that this can be shown to hold for certain growth rates of $p$, also mentioning that it does _not_ hold for $p$ growing with $cN, c\in [0,1]$.  However, for continuous probability distributions, the growth ratio of $p$ is the mass of the probability density function $\mu_{\varepsilon}$ of $\varepsilon$ outside of $\left[ -\frac{1}{2}, \frac{1}{2} \right]$ times $N$, i.e.  $$p = N \cdot \left(  1-\int_{-\frac{1}{2}}^{ \frac{1}{2}} d\mu_{\varepsilon}\right),$$which is either $0$ for probability densities that have all of their support in $\left[ -\frac{1}{2}, \frac{1}{2} \right]$ or else equal to $cN$ with $c \in [0,1]$ (for a standard normal, $c\approx 0.62$, which is pretty high). Thus, practical probability densities such as gaussians fall exactly into the regime of probability densities for which Theorem 1 does not hold, even when the remark is considered.
- The results methodology makes it hard to exactly evaluate how much of the improvements come from the proposed method, as the paper combines Angle-DFQ with a mixed-precision bit allocation (which seems to be hand-tuned, so this might indicate some overfitting). This optimized bit allocation could itself be responsible for a large amount of the gains. Additionally, each network has various different methods used on it (which are not the same for each network), which again use different bit widths each.

### Other
- Large parts of the theory seem only weakly connected to the method, as the theory section is concerned with the special case of ternary quantization (which is nicely solved in close-form), but the method itself is a simple greedy optimization on much larger than ternary grids, that decides between two possible quantization options by brute-force calculating the angle-error

**Questions:**

- I would advise renaming Theorem 1 to Proposition 1 (as the statement as well as the accompanying proof are quite simple).
- While Theorem 1 is very limited in its applicability (see weaknesses section), its general idea seems interesting. Maybe the authors could derive bounds on the angle error for common probability distributions such as gaussians, which would provide a more meaningful contribution
- I would suggest the paper to undergo a major revision of its presentation, using a spell and grammar-checker, using consistent naming and citing (make use of the in-built cite macros from LaTeX), and cleaning up other issues, of which I mentioned some in the weaknesses section. A comparison to some other published and well regarded papers from ICLR or similar conferences might help.
- I would propose to streamline the experiment results by reporting exactly the same configurations of layer-wise bit-widths for each of the methods. Either Angle-DFQ should therefore use uniform bit-widths per layer, or the reported methods should be re-implemented to operate with layer-wise bit allocations. Additionally, the main competitors to Angle-DFQ (which seem to be TNT and Krishnamoorthi, which are also per-layer and data-free) should ideally be reported for each of the networks, if possible.

---

> ### Author Response · Authors · 2024-11-26
>
> Dear respected reviewer,
>
> We would first like to thank the reviewer for their effort. We agree that the presentation could be made clearer. In response to your concerns about the soundness of theorem 1:
>
> In our investigation of angle errors in popular neural networks we found that surprisingly under course quantization schemes the average angle errors in the layers with large dimensions are usually near 90 degrees. We were interested in explaining what conditions create this shocking effect. Theorem 1 offers an explanation. We respectfully disagree that theorem 1 has little relevance to the distributions of real model weights. Under course quantization schemes of large layers in Resnet 18, VGG16, and AlexNet the weights do follow a distribution where most weights are packed around the origin and there are only some outliers outside of the interval x < |½|.  Consider the kernel 163 of the last layer of Resnet 18 after scaling into the integer range but before rounding in the tertiary quantization case. We cannot attach a histogram to this comment due to openReviews policy, but this is easy to verify. Further exploration of the other kernels in this layer would show similar results. We see exactly this behavior of only some fixed number of outliers outside of [-½, ½].
>
> We appreciate the reviewers’ feedback, and we have decided to withdraw the paper to address some issues more comprehensively.

---

### Official Review · Reviewer_pa5L · 2024-10-30

**Soundness:** 2
**Presentation:** 1
**Contribution:** 2
**Rating:** 3
**Confidence:** 4

**Summary:**

This paper proposes to minimize the angle between the pre-trained weights and quantized weights for data-free quantization. The approach consistently outperforms existing state-of-the-art data-free quantization techniques in ImageNet classification.

**Strengths:**

The concept of angle minimization is intriguing, and the experiments demonstrate that it outperforms existing data-free quantization methods.

**Weaknesses:**

1. The significance of the angle between the pre-trained weights and the quantized weights in the context of quantization is unclear. Specifically, why would minimizing the angle help improve the quantization peformance?

2. The algorithm for implementing angle minimization is not well presented.

3. The accuracy achieved on ImageNet does not surpass that of several other state-of-the-art post-training quantization algorithms. With that said, the comparison with other post-training quantization algorithms, such as OBQ [1] and COMQ [2], is inadequate.

4. The reference Zhang et al. (2019) in line 129 is not found in References section.


[1] Frantar, et al., Optimal Brain Compression: A Framework for Accurate Post-Training Quantization and Pruning.

[2] Zhang et al., COMQ: A Backpropagation-Free Algorithm for Post-Training Quantization.

**Questions:**

1. In the proof of Theorem 1, why the limit is equal to 0 in the last step?

---

> ### Author Response · Authors · 2024-11-26
>
> Dear respected reviewer,
>
> We would first like to thank the reviewer for their time. We agree that the presentation could have been better.
>
> In regard to your question:
>
> The limit goes to zero because for n -> infinity the denominator goes to infinity and numerator does not.
>
> In response to your concerns about the weaknesses of the paper::
> 1) The angle error of a weight vector (for example a kernel in an CNN) changes the input space for a positive inner product result, i.e., the ReLU is ‘ON’. This effect is shown in Figure 1. This means that after quantization the set of inputs that turn the Relu ON will be different from the set of inputs that turn the Relu on when the inputs are in full precision.
> 2) Which part of the algorithm presentation was poorly explained from your perspective?
> 3) Our weight rounding method is data-free which is more restrictive than Post-training quantization. For this reason we have claimed the state of the art among data-free approaches.
> 4) Thank you for bringing attention to this error.
>
> We appreciate the reviewers’ feedback, and we have decided to withdraw the paper to address these issues.

---

### Official Review · Reviewer_wUct · 2024-11-01

**Soundness:** 3
**Presentation:** 2
**Contribution:** 3
**Rating:** 5
**Confidence:** 4

**Summary:**

In this paper, the authors propose a **data-free quantization** method that preserves the **direction of weight vectors** rather than simply rounding weights to the nearest quantized value. This approach is applied to each neuron's weights in each layer. Through empirical analysis, they show that traditional nearest-neighbor rounding significantly increases the angle between quantized and original weights, leading to accuracy loss. The authors first derive a mathematical basis for ternary quantization (using \{-1, 0, 1\}) under a Gaussian weight distribution and then introduce a **cosine similarity-based method** that greedily rounds weights to minimize angle error. This technique extends effectively to mixed precision quantization as well. Experiments on popular image classification models demonstrate the method’s effectiveness in preserving model accuracy, especially in low-bit settings.

**Strengths:**

This paper addresses an important research area with practical applications. Its main strengths are:

- Focuses on preserving weight orientation after quantization by minimizing angle deviation, a unique approach not seen in other baselines.
- Data calibration-free, making it suitable for privacy-sensitive applications.
- Enhances low-bit quantization performance.
- Introduces an adaptive threshold specifically for ternary weights.
- Adaptable to mixed precision settings and compatible with other quantization methods.
- Achieves competitive or superior results compared to other data-free quantization approaches.

**Weaknesses:**

While this paper is theoretically sound, both the writing and evaluation have some notable weaknesses:

### Writing Issues:
1. Numerous spelling and grammatical errors, such as "a number or reasons" (line 043), "course weight quantization" (line 152), "The red line demarks the region" (line 184), "asymtotic" (line 323), "out preforms" (line 517), and "bench marked" (line 408), among others.

### Technical Weaknesses:
1. **Computationally Intensive**: The method requires iterative quantization for each layer and neuron, increasing computational demands.
2. **Selective Layer Application for Mixed Precision**: Mixed precision quantization requires careful layer selection, adding complexity.
3. **Limited Applicability to Large Language Models (LLMs)**: The method is untested on LLMs, where quantization is highly relevant, raising questions about its broader practicality.
4. **Narrow Experimental Scope**: The experiments are limited to basic vision tasks, restricting insight into its effectiveness across diverse applications.
5.  Angle-DFQ is applied only to a limited subset of weights.

**Questions:**

1.It is unclear to me whether this method quantized the full weights for each experiment. could you elaborate on that?

2. Is the method computationally intensive, given that each layer requires looping through all neurons? Additionally, does it require storing an optimal scaling constant for each neuron?

3. What criteria are used to select the layers where this method is applied?

4. Does Equation (2) hold in its original form in Algorithm 1, considering that weights are scaled before quantization?

5. Can this method be applied to full large language models (LLMs)?

### Suggestions:

1. I recommend revising the writing for clarity and grammar; using an LLM for assistance may be helpful.

2. In Table 1 It would have been better to show the performance improvement/drop over the reference performance since not all methods have the same reference performance.

3. Extending this method to LLMs or larger models would be valuable, as these applications are more relevant for quantized models.

4. An ablation study could be conducted to evaluate the impact of quantizing all layers versus selected layers and to determine if the choice of scale factors should depend on the architecture or if it can be universal for this method.

---

> ### Author Response · Authors · 2024-11-26
>
> Dear respected reviewer,
>
> We would first like to thank the reviewer for their diligence and hard work. We agree that there is some work to be done on the presentation of the material. In response to your questions:
>
> 1) The time complexity is O(n) and we do not require a scaling constant per neuron, instead for integer quantization the whole layer shares one scale factor.
> 2) The selection criteria is detailed in section 3.5 and refers to theorem 1, we choose the layers that have the highest dimensionality for the angle-DFQ method.
> 3) Yes, imagine a very coarse quantization, tertiary for example.
> 4) Yes, that is a natural extension of this work.
>
> After several concerns raised by the reviewers we have decided to withdraw the paper to address these issues more comprehensively. Thank you for your time and help.

---

### Note · Authors · 2024-11-26

**Comment:**

We appreciate the reviewers’ feedback, and we have decided to withdraw the paper to address some issues in the paper.

**Withdrawal Confirmation:**

I have read and agree with the venue's withdrawal policy on behalf of myself and my co-authors.